# Exploring the Link between Head and Neck Cancer and Elevated Risk of Rheumatoid Arthritis: A National Population-Based Cohort Study

**DOI:** 10.3390/cancers16061081

**Published:** 2024-03-07

**Authors:** Chulho Kim, Hyunjae Yu, Dong-Kyu Kim

**Affiliations:** 1Department of Neurology, Chuncheon Sacred Heart Hospital, Hallym University College of Medicine, Chuncheon 24253, Republic of Korea; gumdol52@hallym.or.kr; 2Institute of New Frontier Research, Division of Big Data and Artificial Intelligence, Hallym University College of Medicine, Chuncheon 24252, Republic of Korea; yunow3618@hallym.or.kr; 3Department of Otorhinolaryngology-Head and Neck Surgery, Chuncheon Sacred Heart Hospital, Hallym University College of Medicine, Chuncheon 24253, Republic of Korea

**Keywords:** cancer, head and neck, rheumatoid, cohort studies, incidence, risk

## Abstract

**Simple Summary:**

Rheumatoid arthritis (RA) exhibits an elevated susceptibility to cancer. Nevertheless, the equivalence of this risk among those afflicted with head and neck cancer (HNC) remains a subject of uncertainty. Our findings unveiled a heightened probability of RA development among individuals with HNC. Notably, this risk is most pronounced immediately following the diagnosis of HNC and exhibits a persistent upward trajectory. Particularly noteworthy is the discerned association between men in their middle age, specifically those with oral cancer, and an augmented linkage to RA. These revelations furnish novel insights and significant information concerning the interconnection between HNC and RA.

**Abstract:**

An increased risk of cancer among patients with rheumatoid arthritis (RA) has been reported. However, the risk of RA events among patients with head and neck cancer (HNC) is unknown. Therefore, we investigated the incidence and risk of RA among patients with HNC. This study was based on a cohort dataset. Overall, 2824 individuals without HNC and 706 patients with HNC were selected using propensity score matching. The overall RA event rate was 12.19 for patients with HNC and 7.60 for those without HNC. A significantly increased risk of developing RA was also observed among patients with HNC. The risk of developing RA over time was relatively high within the first year after HNC diagnosis; further, it increased significantly during the follow-up period. Moreover, middle-aged male patients with HNC exhibited an increased risk of developing RA compared with the controls; however, no significant difference was noted among female patients or other age groups. Notably, subgroup analysis according to cancer subtype revealed that only oral cancer survivors had an increased risk of developing RA. These results underscore the importance of vigilant monitoring by clinicians to promptly identify the onset of RA in patients with HNC.

## 1. Introduction

Rheumatoid arthritis (RA) represents a persistent autoimmune disease characterized by synovial joint inflammation, resulting in manifestations such as pain, swelling, stiffness, and the prospect of consequential joint deformities. Unlike osteoarthritis, which results from the wear and tear of joints, RA involves the immune system mistakenly attacking healthy joint tissues. RA is a multifactorial autoimmune disease influenced by a combination of genetic, environmental, and lifestyle factors [1]. Understanding these risk factors is crucial for identifying individuals with high susceptibility and enhancing preventive measures. Many studies have investigated the relationship between RA and cancer [2,3,4,5,6,7,8]. A possible link between cancer and RA was initially documented in 1978 in a study that revealed an elevated incidence of lymphoma in individuals diagnosed with RA [9]. Recently, a large observational study demonstrated that patients with RA have a heightened risk of developing all types of cancers compared to the general population, indicating a relative excess risk of 20% [6]. Additionally, this study revealed that certain cancers were more prevalent among individuals with RA, including lung, bladder, kidney, upper urinary tract, melanoma, ovarian, cervical, and prostate cancers. There was also an over-representation of certain lymphomas, including diffuse large B-cell lymphoma, follicular lymphoma, Hodgkin’s lymphoma, marginal zone lymphoma, and multiple myeloma [6].

Head and neck cancer (HNC) is a malignant tumor of the head and neck region. These tumors typically originate in areas such as the oral cavity, throat, larynx, pharynx, nasal cavity, paranasal sinuses, and nearby lymph nodes. The common subtypes include oral, pharyngeal, and laryngeal cancers. The advancement of diagnostic technologies in cancer and the refinements in pivotal therapeutic modalities for individuals afflicted with HNC, encompassing surgical interventions, radiation therapy, and chemotherapy, have not only prolonged the overall survival rates of cancer patients but have also concomitantly escalated the frequency of delayed complications. Thus, HNC has exhibited associations with various autoimmune rheumatic diseases [10,11,12]. Moreover, one recent study showed that individuals with autoimmune rheumatic diseases, inclusive of conditions affecting the thyroid, oral cavity, and nasopharynx, exhibited a conspicuously heightened risk of developing HNC in comparison to the general populace [13]. One investigation conducted within the European population revealed an augmented susceptibility to HNC among individuals diagnosed with RA [14]. Nevertheless, to date, there exists a dearth of research scrutinizing the occurrence and risk of RA as a complication in cancer patients, particularly among those afflicted with HNC.

Examining the reciprocal connection between HNC and an increased risk of RA poses challenges owing to the diversity of rheumatic clinical manifestations, various cancer sites susceptible to risk, and exposures related to treatments in both oncology and rheumatology. Hence, within the purview of this investigation, we systematically scrutinized the prevalence and prospective risk of RA among individuals diagnosed with HNC by leveraging a comprehensive dataset representative of the entire national populace. This dataset provides a broad spectrum of diseases, allowing for a comprehensive exploration of the potential interrelations between these specific health conditions. Rigorous control measures were implemented to account for potential confounding variables such as clinical status and demographic factors.

## 2. Materials and Methods

### 2.1. Cohort Dataset and Ethical Considerations

Researchers have constructed a nationwide population-based sample cohort dataset using health insurance claims data collected from the National Health Insurance Service, provided through a de-identification process, and conducted a longitudinal study. Therefore, this study was based on a cohort dataset with a relatively large sample size. As Koreans are compulsorily assigned a unique identification number at birth, medical claims data registered in the cohort dataset cannot be duplicated or missing.

The healthcare provider database also contains information on the type, staffing, and equipment of healthcare facilities. The national database contains information on various medical uses, including the date and time of death, visits to hospitals and outpatient medical facilities, and medication history. The cohort dataset used in this study consisted of a representative sample of 1,025,340 adults, which is approximately 2.2% of the population of the Republic of Korea. Its excellent reliability has already been proven through a reliability verification study and several previous publications [15,16,17,18].

However, the disease diagnosis status in the claims data may not accurately reflect a patient’s health status, depending on the situation. For example, errors may occur when determining whether drug prescriptions are covered by insurance or due to errors or omissions when entering the diagnosis code. Therefore, to improve diagnostic accuracy, careful judgment regarding the use of diagnostic codes is needed among the researcher group.

This investigation received approval from the Institutional Review Board (IRB) at Hallym Medical University, specifically the Chuncheon Sacred Hospital, with the designated protocol code 2021-08-006. Notably, the IRB review process resulted in a waiver of the requirement for written informed consent. This decision was predicated on the absence of identifiable clues or explicit personal information within the cohort database, thereby precluding the need for individual consent. To uphold privacy standards following South Korean regulations, the cohort data were presented to the principal investigator in a de-identified secondary data format.

Due to the stringent policies of the Korean National Health Insurance Service, the datasets produced and scrutinized in this study are not accessible to the public. Nevertheless, the authors affirm that comprehensive data supporting the conclusions drawn in this study are encapsulated within this article. It is imperative to note that this research adhered to the ethical guidelines outlined in the Declaration of Helsinki throughout its execution.

### 2.2. Longitudinal Study Using Retrospective Cohort Design 

To investigate the risk of subsequent RA development in patients with HNC, we selected the matched-target cohort (HNC) and comparison cohort (non-HNC) groups. We compared the incidence and risk rates between the two cohort groups using a retrospective cohort study design. Hence, given the retrospective cohort design employed in this study, all data were sourced from archival records. Observations were initiated from a distinct historical juncture and systematically scrutinized to identify outcomes spanning from that temporal point to the present. Despite the retrospective nature of the study, the analytical framework allowed for the comparison of groups manifesting divergent characteristics concerning specific outcomes that transpired in the past.

The research design was as follows: First, it largely comprised a wash-out period, an index period, and an observation period. For the wash-out period, we selected the first year from the cohort dataset (January–December 2002). This allowed us to eliminate the possibility that RA was diagnosed before HNC. Additionally, we defined the HNC cohort by identifying patients diagnosed with HNC during a specific index period (2003–2005). Participants were selected based on the presence or absence of a diagnostic code indicating HNC. The sub-diagnostic codes encompassing the HNC cohort comprised oral cavity malignancies (C00–C06), salivary gland neoplasms (C07–C08), oropharyngeal carcinomas (C09–C10), nasopharyngeal malignancies (C11–C12), hypopharyngeal tumors (C13–C14), paranasal sinus carcinomas (C30–C31), and laryngeal malignancies (C32).

Concurrently, in the process of assembling a comparison cohort, individuals without HNC were meticulously chosen from the residual cohorts documented in the database. Subsequently, these non-HNC individuals were randomly paired with HNC patients through the application of the propensity score methodology, entailing a fourfold ratio of cancer-free participants for each HNC-afflicted patient. During this matching procedure, variables such as gender, age, residence, income level, and comorbidities were identified as independent variables for achieving equilibrium between the two cohort groups. The quantification of comorbidities was executed through the utilization of the Charlson Comorbidity Index (CCI), a widely employed metric in claims-based research. The CCI is commonly used in research studies and clinical settings to stratify patients based on the comorbidity burden, aiding in predicting outcomes, resource allocation, and treatment decision-making [19,20]. This index is used to predict the ten-year mortality for a patient who may have a range of comorbid conditions. Each condition is assigned a score based on the risk of dying associated with that condition. The scores are then summed to provide a total score that predicts mortality for a patient. The CCI includes a wide array of conditions, including heart disease, AIDS, lung disease, and others. The higher the score is, the more likely the patient’s chances of mortality due to these comorbid conditions. The index has been validated in numerous settings and is widely used in clinical research to adjust for the confounding effects of comorbidities on patient outcomes. It has been adapted for use with administrative health data and has undergone several revisions to update its conditions and scoring system to reflect changes in the understanding and treatment of diseases. The CCI is a valuable tool for researchers and healthcare providers in assessing the burden of comorbidities on patients and predicting their prognoses. The follow-up was terminated if the primary outcome RA; M05, M06 was detected or if the participant died during the observation period. If no specific event occurred in the patient until the final follow-up period in the cohort database, the observation was terminated. The participant selection process is depicted in Figure 1, and the retrospective cohort design is summarized in Figure 2.

### 2.3. Statistical Analysis of Primary Outcomes

The primary objectives of this study encompassed the determination of the incidence and risk ratio of RA in patients afflicted with HNC as compared to those without HNC. The overall RA incidence was computed by dividing the count of RA-diagnosed patients by 1000 person-years, the time frame extending from the patient’s enrollment date to the conclusion of the observational period. To ascertain whether HNC correlated with an elevated hazard ratio for RA development, the Cox proportional hazards regression model was employed. Hazard ratios, denoted as HRs, along with their associated 95% confidence intervals (CIs), were reported both in unweighted and weighted forms. Notably, the Kaplan–Meier method was applied to delineate and juxtapose the cumulative probability of RA onset throughout the entire follow-up duration in the two cohort groups. All statistical analyses were conducted utilizing R software (version 4.0.0), and a *p*-value threshold of <0.05 was established to denote statistical significance.

## 3. Results

### 3.1. Cohort Matching (Comparative Cohort and Target Cohort)

The propensity score method, employing a 4:1 ratio, was employed to equalize the distribution of independent variables between the comparison and target cohorts. Consequently, after the execution of propensity score matching, an examination revealed analogous distributions of all covariates within the two cohort groups. Importantly, statistical analyses confirmed the absence of significant differences in the measured values between the matched cohorts (Table 1). Additionally, a balance-plot test confirmed that similar distributions were observed after matching (Figure 3). Therefore, these findings suggest that the cohort matching between the comparison and target cohorts was appropriate.

### 3.2. Effect of Head and Neck Cancer on Incident Rheumatoid Arthritis Events

In the present investigation, a comprehensive review of 23,829.4 person-years within the comparison cohort and 4841.0 person-years within the target cohort was undertaken to ascertain the incidence and hazard ratios of RA events throughout the follow-up period (Table 2). The findings unveiled a heightened incidence of RA within the HNC patient group, at 12.19 in comparison to the control group at 7.60. Furthermore, the Cox regression analysis delineated a statistically significant weighted risk of 1.49 (95% CI = 1.11–2.01) for RA development in patients with HNC during the follow-up duration.

Similarly, the Kaplan–Meier survival curves, supplemented by log-rank test results, corroborated the higher frequency of RA events among patients with HNC when juxtaposed with the controls. (Figure 4). Analysis of risk-ratio changes over time showed that the risk of developing RA was relatively high during the initial follow-up period after HNC diagnosis (Table 3). Moreover, the weighted risk of the subsequent onset of RA consistently decreased after the first year of HNC diagnosis. However, the hazard ratio for RA events remained consistently significant throughout the follow-up period.

### 3.3. Analysis of Risk for Rheumatoid Arthritis Events According to the Different Subgroups

First, a subgroup analysis according to sex was performed. We observed an increased risk of developing RA in male patients with HNC; however, this difference was not statistically significant (Table 4). Next, subgroup analysis was performed according to age. In our cohort database, the risk of subsequent development of RA was significantly increased in patients with HNC who were aged 45–64 years; however, other age categories showed no statistically significant differences in the weighted hazard ratios (Table 5). Based on these results, we performed a subgroup analysis to evaluate the age-dependent RA risk of male patients with HNC. Interestingly, the weighted hazard ratio for developing RA among male patients with HNC who were aged 45–64 years was 2.62 (95% CI = 1.16–5.90), which showed a statistically significant increase (Table 6). Finally, we analyzed the risk of developing RA according to the HNC subtype. The HNC cohorts were stratified into distinct subtypes, including oral cavity, salivary gland, oropharynx, nasopharynx, hypopharynx, sinus tract, and laryngeal cancer groups. Subsequently, univariate and multivariate Cox regression models were applied to each subtype. The outcomes revealed a statistically noteworthy elevation solely in the weighted hazard ratio within the oral cancer patient group (weighted HR = 1.63, 95% CI = 1.19–2.23). Conversely, no statistically significant hazard ratio alterations were discerned in the remaining HNC subtypes, as delineated in Figure 5.

## 4. Discussion

The association between HNC and RA can be influenced by several factors, including common immunological factors, inflammation, genetic factors, immunomodulatory medications, and behaviorals and environmental factors. Thus, we postulated that the association between cancer and autoimmunity might be bidirectional. To our knowledge, this study constitutes a pioneering endeavor to systematically investigate the risk landscape associated with RA incidence among individuals diagnosed with HNC. By leveraging a robust nationwide population-based cohort dataset, our investigation revealed a discernible proclivity for elevated RA occurrence within the HNC patient cohort compared to their counterparts without HNC. Rigorous adjustment for covariates consistently demonstrated a statistically significant increase in the HR for RA within the HNC population. Temporal scrutiny of the RA risk trajectory revealed a noteworthy surge within the first year after HNC diagnosis, exhibiting sustained elevation and intensification throughout the protracted follow-up period. Intriguingly, the amplified risk of RA manifestation demonstrated statistical significance, specifically among male and middle-aged patients with HNC, in contrast to the absence of significant differences in females or other age strata. A meticulous subgroup analysis predicting distinct cancer subtypes revealed a substantive escalation solely in the risk of RA development among survivors of oral cancer. This refined granularity in our findings highlights the importance of nuanced consideration of cancer subtypes, enriching our understanding of the intricate interdependencies within this specific patient cohort.

The association between cancer and RA remains ambiguous and inadequately elucidated. As our understanding of the underlying mechanisms, long-term consequences, and emergence of new treatments for both cancer and RA continues to progress, the interplay between these conditions becomes more apparent and simultaneously more complex. To date, numerous studies have undertaken investigations into the plausible associations between autoimmune diseases and HNC [2,3,4,5,6,7,8]. These studies reported that various immune and inflammatory pathways could trigger tumorigenesis or that the heightened inflammatory state associated with rheumatic diseases may play a role in initiating and promoting cancer. Conversely, immune responses designed to combat tumors and restrict their growth may inadvertently target self-tissue, leading to the development of autoimmune conditions. Therefore, tumors can function as antigen reservoirs and trigger the initiation of immune responses. Simultaneously, the regenerating cells within specific tissues may act as sources of antigens, thereby contributing to the propagation and maintenance of autoimmunity. Additionally, HNC and RA share a significant risk factor, namely smoking. Previous studies have demonstrated that smoking stands out as the predominant risk factor for the onset of HNC, doubling the likelihood of developing head and neck squamous-cell carcinoma [21,22]. The risk escalates further with an increased intensity of smoking, with a heavier smoking history being correlating with a heightened susceptibility to HNC. Other studies have also reported that smoking could not only elevate the risk of developing RA, particularly among individuals with a genetic predisposition, but could also complicate the treatment of RA if the individual is both a smoker and has the condition [23,24,25]. Thus, the association between smoking and these diseases can be understood through several mechanisms. The shared risk factor of smoking for both HNC and RA highlights the complex interplay between environmental exposures, genetic predisposition, and immune system dysfunction. While the direct mechanisms may differ—carcinogenesis in the case of HNC and autoimmunity in RA—the systemic effects of smoking on inflammation, immune modulation, and cellular damage contribute to the increased risk of both conditions [26,27,28]. Understanding these mechanisms underscores the importance of smoking cessation and prevention strategies in reducing the incidence and severity of both head and neck cancer and rheumatoid arthritis, alongside other smoking-related diseases.

As it is challenging to accurately assess infrequent events such as cancer within the context of rheumatic diseases, extensive longitudinal epidemiological studies offer a valuable opportunity to explore the connections between malignancy and autoimmunity. Additionally, the intricate relationship of cancer with specific autoimmune conditions is subject to a myriad of complex and diverse influences. Thus, large-scale studies could provide an excellent opportunity to gain insight into the intricate relationship between autoimmune conditions and cancer development. Notably, a recent systematic review and meta-analysis revealed a 2.35-fold increase in the overall incidence of HNC among individuals with autoimmune rheumatic diseases, surpassing the rates observed in the general population [13]. Moreover, the present study identified a significant 1.49-fold increase in the overall incidence of RA events among patients with HNC compared to controls. Collectively, these findings provide robust epidemiological evidence supporting the hypothesis of a bidirectional relationship between HNC and RA.

Notably, previous longitudinal studies have shown different risks based on sex. One retrospective cohort study revealed that men exhibited significantly elevated risks of lung, liver, and esophageal cancers while experiencing a reduced risk of prostate cancer, whereas women demonstrated a notable decrease in the risk of several cancers, including breast, ovarian, uterine, cervical, and melanoma, with a risk reduction compared with the general population [29]. Another extensive observational study demonstrated that an elevated risk was predominantly observed among males, manifesting as a 34% relative increase in the overall cancer risk, whereas females exhibited a comparatively modest increase in relative risk, totaling 8% when compared to the risk observed in the general population [6]. Similarly, our investigation revealed a higher incidence and risk of RA development among male patients with HNC. Additionally, we found the highest incidence and risk of RA among patients aged 45–64 years.

Our study has several notable strengths. First, this was an inaugural cohort analysis utilizing nationwide population-based data to scrutinize the subsequent emergence of RA in individuals diagnosed with HNC. Employing the propensity score matching method, we effectively controlled for crucial confounding variables, facilitating a rigorous comparison of RA incidence between the HNC and matched control groups. While refraining from making definitive causal inferences, our findings offer valuable insights and implications for clinical practice, prompting the consideration of potential associations between HNC and RA. Second, our study had a robust design, encompassing a large patient cohort and an extensive 11-year observation period. This prolonged timeframe allowed a comprehensive examination of the temporal relationship between HNC and RA development. Third, endeavors were undertaken to attenuate surveillance bias in the evaluation of RA risk among patients diagnosed with HNC. This was achieved by meticulously selecting sociodemographically matched controls from a cohort database. By ensuring that the control group shared similar sociodemographic characteristics with the patients with HNC, this study aimed to reduce potential biases related to surveillance practices. This approach enhanced the reliability of the findings and provided a more accurate evaluation of the association between HNC and the risk of developing RA. Fourth, RA is a disease that requires time for patients to visit the hospital and receive an accurate diagnosis, even when symptoms appear. As well as the possibility of underdiagnosed RA, there is always a length-of-time bias in which RA may have existed before patients with cancer were diagnosed. Thus, to remove this, we set a wash-out period of 1 year before the index period. This means that our research design minimized length-of-time bias. Finally, we systematically assessed variations in the risk dynamics of RA development with increasing observation periods. This study aimed to determine whether the observed risk of RA in individuals with HNC was a coincidental occurrence confined to specific temporal intervals or manifested as a sustained phenomenon following a discernible pattern. Our findings revealed a sustained increase in the long-term risk of developing RA after HNC diagnosis. Similar to previous reports [30,31,32], our findings suggest a substantive association between HNC and RA occurrence, indicating a non-random correlation rather than a chance event.

However, it is crucial to delineate the limitations of this study and approach its findings judiciously. Firstly, the identification of diseases relied upon diagnostic codes from the ICD-10-CM system, lacking the granularity provided by detailed medical records. Additionally, our investigation has omitted essential information such as comprehensive medical histories and pathological reports. Thus, our cohort database included not only squamous-cell carcinoma but also non-squamous-cell carcinoma. Due to these limitations, we have a plan for future prospective clinical studies, incorporating a more comprehensive array of variables, to be conducted to delineate the complex pathophysiological mechanisms underlying the correlation between these conditions. Second, we used a cohort sample database with a restricted set of identifiable variables. This limitation prevented the correction for some risk factors that could potentially influence the occurrence of RA, including family history and smoking. Third, due to the unavailability of information on treatment methods for HNC, this study could not assess the impact of cancer treatments, such as chemotherapy and radiotherapy, on the risk of developing RA. Comprehending the outcomes of these therapeutic interventions is pivotal for elucidating the intricate interplay between oncological disorders and autoimmune diseases. Such an understanding could illuminate potential shared pathogenic pathways, reveal implications for prognosis, and inform the development of targeted treatment strategies that address both the primary condition and its immunological consequences. Regrettably, the database did not encompass information on treatment modalities. We acknowledge this as a significant limitation and will endeavor to address it in subsequent research endeavors. Fourth, the age data in the database were presented in groups owing to de-identification concerns. Consequently, the matching of the two groups was based on categorized age data rather than on the actual age distribution, potentially introducing residual bias into the study analysis. Additionally, our cohort group may have a potential discrepancy between the time of diagnosis and the actual onset of the disease across patients. To overcome this issue, we have made concerted efforts to align variables such as access to healthcare facilities and sociodemographic characteristics (residence, income, age, gender, etc.) between the groups, thereby striving to minimize any biases and ensure a more balanced analysis. Finally, the retrospective cohort design of the study limited direct investigation and analysis of the pathological mechanisms underlying HNC and RA. Overall, while this study contributes valuable insights, the acknowledged limitations underscore the exigency for subsequent research endeavors to furnish a more nuanced comprehension of the interrelationship between HNC and RA.

## 5. Conclusions

This investigation revealed an elevated susceptibility to RA events among individuals diagnosed with HNC. Notably, the risk of developing RA was promptly observed post-HNC diagnosis, with a sustained and statistically significant increase over subsequent periods. Particularly noteworthy was the heightened association with RA events discerned among male individuals within the middle-aged demographic and among those afflicted with oral cavity cancer. These findings contribute novel and substantive insights into the intricate interplay between HNC and RA. Consequently, our investigation suggests that clinicians should be advised to exercise heightened vigilance regarding the potential onset of RA in the care of HNC patients and to implement precautionary measures facilitating early diagnostic interventions.

## Figures and Tables

**Figure 1 cancers-16-01081-f001:**
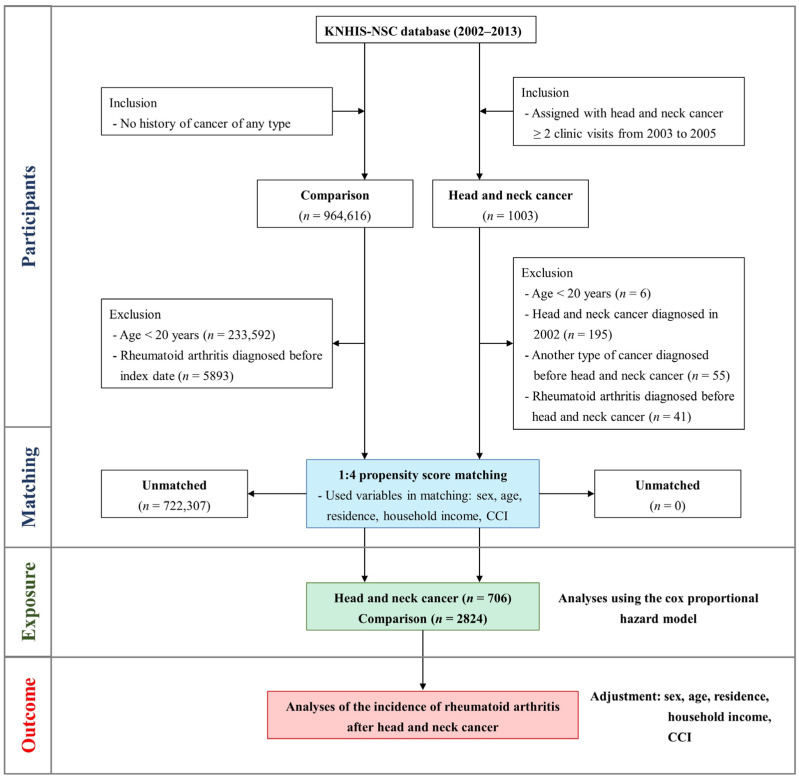
Brief illustration of the process of participant selection in the present study.

**Figure 2 cancers-16-01081-f002:**
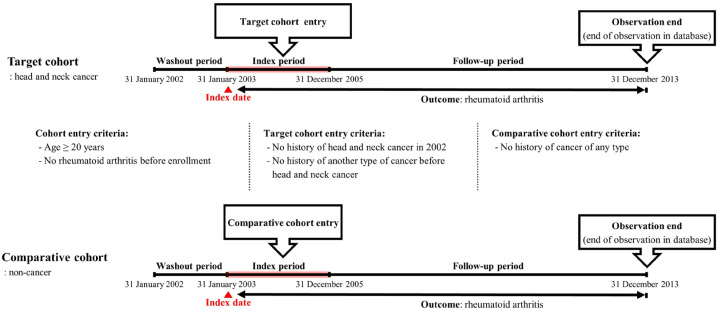
Description of the longitudinal study design (retrospective cohort fashion).

**Figure 3 cancers-16-01081-f003:**
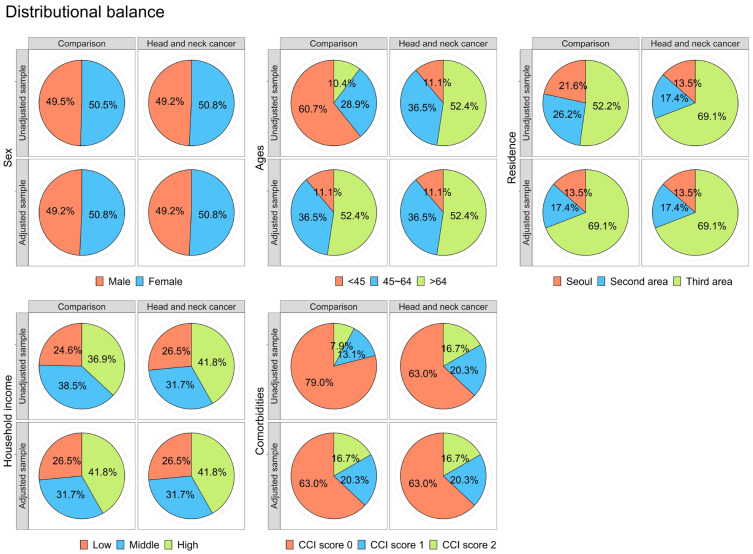
Cohort alignment through the utilization of a balanced plot to assess equilibrium across selected covariables.

**Figure 4 cancers-16-01081-f004:**
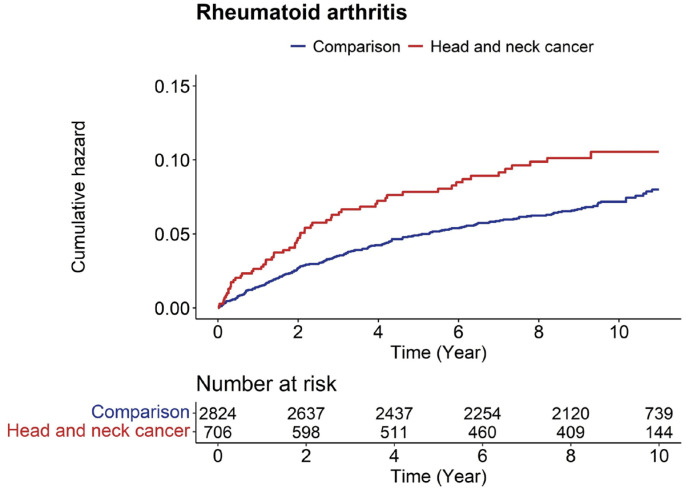
The cumulative hazard ratio of rheumatoid arthritis within the two groups was assessed relative to the progression of the follow-up period.

**Figure 5 cancers-16-01081-f005:**
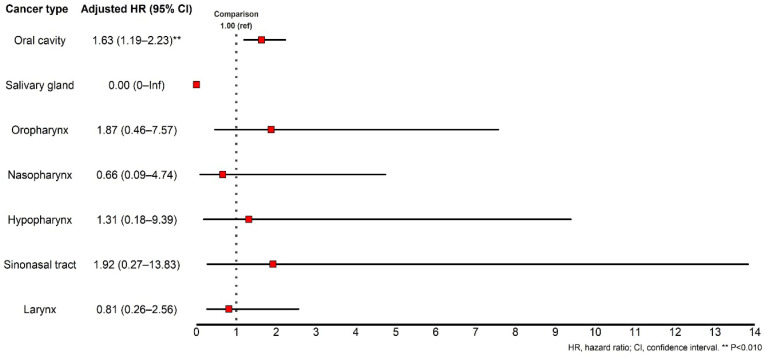
Hazard ratio visualization for incident rheumatoid arthritis diagnosis according to the specific subtypes of head and neck cancer.

**Table 1 cancers-16-01081-t001:** Demographic and clinical profiles of the study cohort.

Variables	Comparative Cohort (*n* = 2824)	Target Cohort (*n* = 706)	*p*-Value
Sex			1.000
Male	1388 (49.2%)	347 (49.2%)	
Female	1436 (50.8%)	359 (50.8%)	
Age (years)			1.000
<45	312 (11.0%)	78 (11.0%)	
45–64	1032 (36.5%)	258 (36.5%)	
>64	1480 (52.4%)	370 (52.4%)	
Residence			1.000
Seoul	380 (13.5%)	95 (13.5%)	
Second area	492 (17.4%)	123 (17.4%)	
Third area	1952 (69.1%)	488 (69.1%)	
Household income			1.000
Low (0–30%)	748 (26.5%)	187 (26.5%)	
Middle (30–70%)	896 (31.7%)	224 (31.7%)	
High (70–100%)	1180 (41.8%)	295 (41.8%)	
CCI			1.000
0	1780 (63.0%)	445 (63.0%)	
1	572 (20.3%)	143 (20.3%)	
≥2	472 (16.7%)	118 (16.7%)	

Seoul, largest metropolitan area; second area, other metropolitan cities; third area, other areas; CCI, Charlson Comorbidity Index.

**Table 2 cancers-16-01081-t002:** The investigation focused on evaluating the incidence rate and risk ratio associated with the occurrence of incident rheumatoid arthritis events during the follow-up period.

Variables	N	Case	Person-Year	Incidence Rate	Unweighted HR (95% CI)	Weighted HR (95% CI)
Incident rheumatoid arthritis events						
Comparative cohort	2824	181	23,829.4	7.60	1.00 (ref)	1.00 (ref)
Target cohort	706	59	4841.0	12.19	1.52 (1.13–2.04) **	1.49 (1.11–2.01) **

HR, hazard ratio; CI, confidence interval. ** *p* < 0.010.

**Table 3 cancers-16-01081-t003:** Elucidation of the temporal dynamics of the risk of rheumatoid arthritis events, specifically with the elapsed time since the diagnosis of head and neck cancer.

Time (Year)	Rheumatoid Arthritis	Rheumatoid Arthritis
	Unweighted HR (95% CI)	Weighted HR (95% CI)
1	1.89 (1.08–3.31) *	1.90 (1.09–3.32) *
2	1.78 (1.17–2.72) **	1.79 (1.17–2.72) **
3	1.80 (1.24–2.60) **	1.80 (1.24–2.60) **
4	1.73 (1.22–2.44) **	1.72 (1.22–2.43) **
5	1.63 (1.17–2.27) **	1.62 (1.16–2.25) **
6	1.61 (1.17–2.22) **	1.59 (1.16–2.20) **
7	1.59 (1.17–2.18) **	1.57 (1.15–2.15) **
8	1.61 (1.19–2.18) **	1.59 (1.17–2.15) **
9	1.57 (1.16–2.12) **	1.55 (1.15–2.08) **
10	1.55 (1.15–2.08) **	1.52 (1.13–2.04) **
11	1.52 (1.13–2.04) **	1.49 (1.11–2.01) **

HR, hazard ratio; CI, confidence interval. * *p* < 0.05, ** *p* < 0.010.

**Table 4 cancers-16-01081-t004:** Examination of the risk ratios of rheumatoid arthritis delineated by gender between two cohorts.

Sex	Male		Female	
	Comparison	HNC	Comparison	HNC
Rheumatoid arthritis				
Unweighted HR (95% CI)	1.00 (ref)	1.82 (1.07–3.08) *	1.00 (ref)	1.36 (0.95–1.94)
Weighted HR (95% CI)	1.00 (ref)	1.84 (1.08–3.12) *	1.00 (ref)	1.38 (0.96–1.96)

HNC, head and neck cancer; HR, hazard ratio; CI, confidence interval. * *p* < 0.05

**Table 5 cancers-16-01081-t005:** Examination of the risk ratios of rheumatoid arthritis delineated by age between two cohorts.

Age	<45		45–64		>64	
	Comparison	HNC	Comparison	HNC	Comparison	HNC
Rheumatoid arthritis						
Unweighted HR (95% CI)	1.00 (ref)	0.50 (0.06–3.94)	1.00 (ref)	2.04 (1.35–3.09) ***	1.00 (ref)	1.24 (0.81–1.92)
Weighted HR (95% CI)	1.00 (ref)	0.43 (0.05–3.40)	1.00 (ref)	2.01 (1.33–3.05) ***	1.00 (ref)	1.22 (0.79–1.88)

HNC, head and neck cancer; HR, hazard ratio; CI, confidence interval. *** *p* < 0.001.

**Table 6 cancers-16-01081-t006:** Examination of the risk ratios of rheumatoid arthritis delineated according to the age group of male individuals between two cohorts.

Age	<45		45–64		>64	
	Comparison	HNC	Comparison	HNC	Comparison	HNC
Rheumatoid arthritis						
Unweighted HR (95% CI)	1.00 (ref)	-	1.00 (ref)	2.63 (1.17–5.94) *	1.00 (ref)	1.48 (0.73–3.00)
Weighted HR (95% CI)	1.00 (ref)	-	1.00 (ref)	2.62 (1.16–5.90) *	1.00 (ref)	1.46 (0.72–2.96)

HNC, head and neck cancer; HR, hazard ratio; CI, confidence interval. * *p* < 0.05.

## Data Availability

The datasets generated and analyzed in the current study are not publicly available owing to the policy of the Korea National Health Insurance Service, but they are available from the corresponding author upon reasonable request.

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
