# Peer review of "Exploring the Link between Head and Neck Cancer and Elevated Risk of Rheumatoid Arthritis: A National Population-Based Cohort Study"

_cancers, 2024, doi:10.3390/cancers16061081_

Round 1

Reviewer 1 Report

Comments and Suggestions for Authors

I have reviewed this interesting retrospective national population-bassed cohort study about the link between head and neck cancer (HNC) and rheumatoid arthritis (RA).

There is a heightened risk of patients with RA develop all types of cancers compared to the general population, indicating a relative excess risk of 20%. Some studies demonstrated that HNC has exhibit associations with autoimmune rheumatic diseases and vice-versa. The findings of the present study are confirmatory in a large cohort with adequate design.

Introduction, line 37: please replace “pathology” by disease or condition.

Introduction: The author must mention that they refer to squamous cell carcinoma (SCC).

Methods, 2.2.: Were cases of non-SCC HNS included?

Discussion, line 288 and the following ones: smoking is a significant risk factor for both HNC and RA. It is not clearly demonstrated in the Discussion.

Author Response

Introduction, line 37: please replace “pathology” by disease or condition.

Introduction: The author must mention that they refer to squamous cell carcinoma (SCC).

à Response: First of all, we sincerely appreciate the evaluation of the referees. We agree with the reviewer’s advice. Thus, as suggested, we modified it. In this investigation, we were not furnished with data to delineate the histologic distinctions among cancer patients. Consequently, in addition to SCC, the HNC cohort also encompasses NSCC.

Methods, 2.2.: Were cases of non-SCC HNS included?

.à Response: In this study, specific histologic characteristics distinguishing cancer patients were not available. Therefore, the cohort of Head and Neck Cancer (HNC) included not only Squamous Cell Carcinoma (SCC) but also Non-Squamous Cell Carcinoma (NSCC). This limitation was elaborated upon in the manuscript's discussion section.

Discussion, line 288 and the following ones: smoking is a significant risk factor for both HNC and RA. It is not clearly demonstrated in the Discussion.

à Response: We agree with the reviewer’s advice. As you commented, we added more description as follows: “The association between smoking and these diseases can be understood through several mechanisms. The shared risk factor of smoking for both HNC and RA highlights the complex interplay between environmental exposures, genetic predisposition, and immune system dysfunction. While the direct mechanisms may differ—carcinogenesis in the case of HNC and autoimmunity in RA—the systemic effects of smoking on inflammation, immune modulation, and cellular damage contribute to the increased risk of both conditions. Understanding these mechanisms underscores the importance of smoking cessation and prevention strategies in reducing the incidence and severity of both head and neck cancer and rheumatoid arthritis, alongside other smoking-related diseases.”

Reviewer 2 Report

Comments and Suggestions for Authors

Thank you for the opportunity review this research article.  Kim et al presents a novel paradigm of conducting population level study using retrospective data to discover associations between cancer and rheumatoid arthritis

While the methodology is considered and well presented, it is important to not overstate the associations discovered during this retrospective data analysis.

Head and neck cancer is also a very heterogeneous group of malignancies, while an attempt to categorise based on anatomical boundaries have been made, histopathological differentiation is not made.  If the data is available, this would likely inform a causal relationship to why certain anatomical subtypes are more closely associated with RA compared to others.

Treatment modality may also have a causal relationship to RA after a diagnosis of HNC, again if the data is available this may improve your results and discussion.

While the groups were controlled for absence of RA prior to the recruitment phase, was other auto-immune conditions or risk factors for RA used in the propensity score matching?  -  because your data suggests a very significant rate of RA in HNC group compared to the standard population, which I have not observed in my practice in Australia.  May need to further check whether this is a Korean phenomenon or if reported elsewhere also?

Comments on the Quality of English Language

Well written

Author Response

Head and neck cancer is also a very heterogeneous group of malignancies, while an attempt to categorise based on anatomical boundaries have been made, histopathological differentiation is not made.  If the data is available, this would likely inform a causal relationship to why certain anatomical subtypes are more closely associated with RA compared to others.

à Response: First of all, we sincerely appreciate the evaluation of the referees. We agree with the reviewer’s opinion. Unfortunately, specific histologic characteristics distinguishing cancer patients were not available in the database used in the present study. Therefore, the cohort of Head and Neck Cancer (HNC) included not only Squamous Cell Carcinoma (SCC) but also Non-Squamous Cell Carcinoma (NSCC). This limitation was elaborated upon in the manuscript's discussion section.

Treatment modality may also have a causal relationship to RA after a diagnosis of HNC, again if the data is available this may improve your results and discussion.

à Response: Regrettably, the database did not encompass information on treatment modalities. We acknowledge this as a significant limitation and will endeavor to address it in subsequent research endeavors. We kindly request your understanding regarding these constraints, which we have elaborated upon with greater detail in the discussion section of our manuscript.

While the groups were controlled for absence of RA prior to the recruitment phase, was other auto-immune conditions or risk factors for RA used in the propensity score matching?  -  because your data suggests a very significant rate of RA in HNC group compared to the standard population, which I have not observed in my practice in Australia.  May need to further check whether this is a Korean phenomenon or if reported elsewhere also?

à Response: Thank you for pointing out a very important point. In this study, the presence of comorbidities was corrected using CCI, which is mainly used in claims data research. Therefore, the incidence of comorbidities (e.g. autoimmune diseases, etc.) was controlled to be the same between the two groups (HNC and non-HNC group). However, smoking history, which is another major risk factor, could not be controlled. Additionally, as described as a limitation, differences in treatment methods (e.g., use of anticancer drugs, etc.) and histological characteristics of carcinoma are not reflected, so further study is required to accurately determine whether the results of the present study are a coincidence or a meaningful causal relationship.

Reviewer 3 Report

Comments and Suggestions for Authors

The reviewed study deals with an exploration of the link between HNC and rheumatoid arthritis. Throughout the study starting from  Introduction and further in the studies description as well as in Discussion an uncertainty appears  if HNC increases risk to develop RA or vice versa. Too much extent the authors are trying to turn reader’s attention on immune system deregulation in both diseases. It could be recognized as contributing factor or a consequence of disease. Reader is not getting a clear answer.

Definitively the authors have shown an elevated risk to develop RA in patients with HNC. The most pronounced association was seen in middle-ages male subjects. By the way they are a highest risk group to develop HNC and within it laryngeal cancer. A good starting point to carry out studies was assignment of unique identification number to Koreans that further n appease in the medical documentation.

I am having doubt about using HNC diagnosis time as a threshold of risk. Diagnosis time is fairly individual. Next a comparison was done =between HNC and non-HNC subjects potentially evolving RA. Did you check new CA cases in non-cancer group? Smoking/drinking habits a strong causative agent in HNS (laryngeal) cancer was not taken into account.

Minor remarksŁ

line 95, parenthesis missing.

Insufficient bibliographic information at references: 4,15,  16, 17 and 22.

Author Response

The reviewed study deals with an exploration of the link between HNC and rheumatoid arthritis. Throughout the study starting from Introduction and further in the studies description as well as in Discussion an uncertainty appears if HNC increases risk to develop RA or vice versa. Too much extent the authors are trying to turn reader’s attention on immune system deregulation in both diseases. It could be recognized as contributing factor or a consequence of disease. Reader is not getting a clear answer.

à Response: First of all, we sincerely appreciate the evaluation of the referees. We agree with the reviewer’s opinion. This investigation adopts the form of a population-based cohort study, which inherently limits our ability to discern if the outcomes observed are the result of temporal associations or derive from a definitive causal linkage. Nevertheless, we thought further subgroup analysis suggests a strong likelihood of an underlying connection between the two conditions, rather than mere coincidental occurrences. If the reviewer understands the limitations of the current study, we will present research data that overcomes the limitations of this study through additional research. Thank you for taking your valuable time to write a review.

Definitively the authors have shown an elevated risk to develop RA in patients with HNC. The most pronounced association was seen in middle-ages male subjects. By the way they are a highest risk group to develop HNC and within it laryngeal cancer. A good starting point to carry out studies was assignment of unique identification number to Koreans that further n appease in the medical documentation. I am having doubt about using HNC diagnosis time as a threshold of risk. Diagnosis time is fairly individual. à Response: The reviewer rightly notes the potential discrepancy between the time of diagnosis and the actual onset of the disease across patients. It's important to recognize that, within the constraints of our study, using the time of diagnosis as the marker for disease occurrence is our most viable method. To address concerns related to the timing of diagnosis, we have made concerted efforts to align variables such as access to healthcare facilities and sociodemographic characteristics (residence, income, age, gender, etc.) between the groups, thereby striving to minimize any biases and ensure a more balanced analysis.

Next a comparison was done =between HNC and non-HNC subjects potentially evolving RA. Did you check new CA cases in non-cancer group? Smoking/drinking habits a strong causative agent in HNS (laryngeal) cancer was not taken into account.

à Response: In this investigation, we evaluated the risk of developing Rheumatoid Arthritis (RA) by comparing individuals diagnosed with Head and Neck Cancer (HNC) to those without such diagnoses. To ensure a fair comparison regarding patients' pre-existing health conditions, we matched the two groups based on the Charlson Comorbidity Index (CCI), which is the standard approach for handling comorbidities in studies utilizing claims data. It is important to note, however, that our study's significant limitation lies in the unavailability of data concerning participants' smoking and alcohol use habits within the cohort database, preventing us from adjusting for these potentially influential factors. This limitation has been thoroughly discussed in the manuscript to maintain transparency and acknowledge the potential impact on our study's conclusions.

Minor remarks

line 95, parenthesis missing.

Insufficient bibliographic information at references: 4,15, 16, 17 and 22.

à Response: Thank you for your comment. We modified these errors.